# Analysis of the Whole-Genome Sequences from an *Equus* Parent-Offspring Trio Provides Insight into the Genomic Incompatibilities in the Hybrid Mule

**DOI:** 10.3390/genes13122188

**Published:** 2022-11-23

**Authors:** Xiujuan Ren, Yuanyi Liu, Yiping Zhao, Bei Li, Dongyi Bai, Gerelchimeg Bou, Xinzhuang Zhang, Ming Du, Xisheng Wang, Tugeqin Bou, Yingchao Shen, Manglai Dugarjaviin

**Affiliations:** Equine Research Center, College of Animal Science, Inner Mongolia Agricultural University, 306 Zhaowuda Road, Hohhot 010018, China

**Keywords:** *equus*, heterogeneous hybridization, genomic incompatibility, Mendelian inheritance error

## Abstract

Interspecific hybridization often shows negative effects on hybrids. However, only a few multicellular species, limited to a handful of plants and animals, have shown partial genetic mechanisms by which hybridization leads to low fitness in hybrids. Here, to explore the outcome of combining the two genomes of a horse and donkey, we analyzed the whole-genome sequences from an *Equus* parent-offspring trio using Illumina platforms. We generated 41.39× and 46.21× coverage sequences for the horse and mule, respectively. For the donkey, a 40.38× coverage sequence was generated and stored in our laboratory. Approximately 24.86 million alleles were discovered that varied from the reference genome. Single nucleotide polymorphisms were used as polymorphic markers for assigning alleles to their parental genomic inheritance. We identified 25,703 Mendelian inheritance error single nucleotide polymorphisms in the mule genome that were not inherited from the parents through Mendelian inheritance. A total of 555 *de novo* single nucleotide polymorphisms were also identified. The rate of *de novo* single nucleotide polymorphisms was 2.21 × 10^−7^ in the mule from the *Equus* parent-offspring trio. This rate is obviously higher than the natural mutation rate for *Equus*, which is also consistent with the previous hypothesis that interracial crosses may have a high mutation rate. The genes associated with these single nucleotide polymorphisms are mainly involved in immune processes, DNA repair, and cancer processes. The results of the analysis of three genomes from an *Equus* parent-offspring trio improved our knowledge of the consequences of the integration of parental genomes in mules.

## 1. Introduction

The caballine lineage and the stenonine lineage diverged quite recently, approximately 4.0–4.5 million years ago [1]. The earliest occurrence with gene flow was about 2.1–3.4 million years ago [2]. *Equus* speciation events were accomplished through acute chromosomal rearrangements, with the rearrangement rate ranging from 2.9 to 22.2 per million years [3,4]. Two sets of perfectly functional and structurally well-characterized genetic programs have evolved for horses (*Equus caballus*, 2n = 64) and donkeys (*Equus asinus*, 2n = 62) [5,6,7]. Although the crossing of horse and donkey is capable of producing offspring, the mule (2n = 63, offspring of male donkey and female horse) and hinny (2n = 63, offspring of male horse and female donkey) are scientifically considered to be incapable of natural mating to produce offspring.

According to Dobzhansky-Muller incompatibilities (DMIs), the evolutionary accumulation of mutational differences between species is attributed to the conflict and coevolution among and within the genomes of the organisms, which are deleterious genetic loci in their hybrid offspring [8,9]. Interactions between such two or more alleles left behind by parents will reduce the fitness of interspecific or interpopulation hybrids, including hybrid sterility, inviability, lethality and weakness [10]. For young species or incomplete speciation, these disruptive interaction loci might arise at single genes evolving divergently across species through positive selection [11], at duplicate genes losing function in different paralogues from diverging populations [12], or at inversions and translocations [13,14], which ultimately drive speciation by causing intrinsic postzygotic reproductive isolation. For older species, such as the horse and donkey, except for the aforementioned detrimental positions driving speciation, more complex genetic incompatibilities have continuously accumulated across the genomes of two divergent species. We refer to these incompatibilities as generalized DMIs, which will produce interaction dysfunction when brought together in a hybrid background. It is possible that genes of the immune system have played important roles in generalized DMIs in vertebrates. Major histocompatibility complex (*MHC*) genes are characterized as trans-species polymorphisms maintained by pathogen-driven balanced selection [15,16]. Several studies in mice [17] and teleost fish [18] have implicated that these interspecific polymorphisms in the *MHC* genes will reduce the fitness of the F1 hybrids. Recent work about hybrid necrosis in three species of *Capsella* also showed that the polymorphisms fixed by balancing selection might become interspecific barriers [19]. Immunoglobulin genes (IGs) are not conserved across the closely related sister taxa, driven by the diversity of target microorganisms for IG responses [20]. The integration of divergent IG loci from *Mus musculus* subspecies into their F1 hybrids results in an apparent mismatch of heavy- and light-chain loci and reduced fitness, which predisposes them to autoimmune complications [21]. Allelic divergences between species may disrupt interactions between their products, such as parental regulatory divergences. This commonly results in hybrid misexpression, which will reduce the survival of offspring F1 [22]. Interspecific hybridization between two species of swordtail fish (e.g., *Xiphophorus birchmanni* × *Xiphophorus malinche* and *Xiphophorus maculatus* × *Xiphophorus hellerii*) causes melanoma in their hybrids [23]. This hybrid incompatibility is the result of an interaction between the melanoma receptor tyrosine-protein kinase (*xmrk*) gene and an unknown locus [24]. In addition to hybrid lethality and sterility, lethal melanocyte tumorigenesis in swordtail fish hybrids is an innovative mechanism for decreasing hybrid fitness.

Under this deleterious mutational stress during early embryonic development, the molecular mechanism by which heterozygotes from divergent species protect the survival of the F1 hybrids from self-correcting rejection remains unknown. Several studies have provided strong evidence that somatic mutations occur post-zygotically during early embryogenesis. These mutations are recognized as important raw materials for genetic diversity. However, they are generally considered to be a prominent cause of disease in normal individuals, and such diseases include cancer [25], autism [26], and some rare developmental disorders [27]. Nevertheless, some hybrids still survive their frequently mutated genomes, such as the high rate of offspring-specific mutations detected in the F1 hybrids of the goldfish × common carp cross [28]. Previous whole-genome mutation rate studies in plants showed that highly heterozygous lines have a higher mutation rate than homozygous lines. For example, there is a 3.6-fold higher rate in heterozygous thale cress, a 3.4-fold higher rate in heterozygous rice, and a more modest 1.6-fold increase in a hybrid peach tree (*Prunus davidiana* × *Prunus persica*) versus in a weakly heterozygous peach tree (*P. persica*) [29,30]. These early postzygotic mutations may reflect hidden genetic incompatibilities ascribed to evolutionarily accumulated mutational differences between species. However, whether these rapidly mutating single nucleotide polymorphisms (SNPs), triggered by interspecific hybridization stress, can buffer the imbalance between parental haploid genomes and thus protect the survival of hybrid individuals remains unknown.

Combining previous reports, more than 98% of the horse genome conserved sequence was covered (up to five depth) by the donkey genome data. (Appendix A) [5,31]. The high similarity exposes one of the biggest challenges in sequencing the mule genome, which is the difficulty in discriminating homologous sequences inherited from progenitors. This limits the future analysis of the outcomes of combining two genomes. In this study, we sequenced the whole-genome sequences from an *Equus* parent-offspring trio using the Illumina platform. Family-based sequencing is powerful in analyzing the inheritance patterns of genotypes. Guided by SNP markers, we assigned the alleles to their genomic inheritance and identified non-Mendelian SNPs and *de novo* SNPs in the mule genome. Through functional analysis of these rapidly mutating SNPs, we assessed the impact of hybridization on the survival of the mules.

## 2. Materials and Methods

### 2.1. Samples for Genomic DNA

Whole-genome sequences were characterized from three members of an *Equus* parent-offspring trio, consisting of a female horse, a male donkey and their hybrid offspring (a female mule). The three animals originated from the Xilingol League of Inner Mongolia, China. Data for donkey genome sequences used in this study was generated and stored in our laboratory [5]. For the horse and the mule, approximately 5 mL of peripheral blood was collected for DNA extraction. Blood samples were collected during veterinary examinations. No animal was hurt or captured as a result of these studies.

### 2.2. Genome Sequences

For the horse and the mule, DNA was extracted from peripheral blood cells. PE libraries were sequenced using the Illumina HiSeq X-ten (2 × 150 bp). Standard genomic library preparation and sequencing followed the manufacturer’s instructions, and sequence reads were collected from the Illumina data processing pipeline.

### 2.3. Data Filtering

AdapterRemoval (version 2.0, Centre for GeoGenetics, Natural History Museum of Denmark, University of Copenhagen, Copenhagen, Denmark) [32] was used to trim adapter sequences from sequence reads generated by Illumina HiSeq X-ten. Low-quality sequences were defined using sliding windows of 5 bp with a step size of 1 bp. If the average quality value (Q) was <20 for five consecutive bases or the Phred quality score was ≤2 for the last base, we trimmed reads from the last base in the windows. Finally, only high-quality PE reads with ≥50 nucleotides were selected.

### 2.4. Mapping Reads to the Thoroughbred Horse Reference Sequence

High-quality PE reads from the horse, donkey and mule were mapped to the reference genome sequence of the thoroughbred horse (*Equcab 3.0*) using BWA (version 0.7.5a-r416, Wellcome Trust Sanger Institute, Wellcome Trust Genome Campus, Cambridge, UK) software with default parameters [33]. Unique alignments were generated using the SAMtools (Version 0.1.19-44428cd, Wellcome Trust Sanger Institute, Wellcome Trust Genome Campus, Cambridge, UK) package with the parameter options “-q 30” [34].

### 2.5. SNP Calling

The unique alignments were used for calling SNPs and Indels. SNPs were detected by comparison of mapped sequences between the reference genome and each sample. Loci with heterozygous and homozygous genotypes in each sample but different from the reference bases were detected as SNPs. Picard (version 1.93, The Broad Institute of Harvard and MIT, Cambridge, MA, USA) software was used to mark potential duplicates. SAMtools and GATK (version 3.5-0-g36282e4, Program in Medical and Population Genetics, The Broad Institute of Harvard and MIT, Cambridge, MA, USA) [35] were used to call SNPs for each individual separately, and only the intersection of SNPs identified by the two software programs was used for subsequent analysis [36]. The HaplotypeCaller program of GATK was used to obtain SNPs with the parameters: -stand_call_conf 30 and -stand_emit_conf 10. To obtain the high-confidence variants, the initially obtained candidate sites were strictly filtrated with the following parameters: --clusterSize 3 --clusterWindowSize 10, --filterExpression “QUAL < 30.0”, --filterExpression “QD < 2.0”, --filterName “FilterFS” --filterExpression “FS > 20.0”, --filterName “FilterMQ” --filterExpression “MQ < 20.0”, --filterName “FilterMQRankSum”, --filterExpression “MQRankSum < -3.0”, --filterName “FilterReadPosRankSum”, --filterExpression “ReadPosRankSum < -3.0”, --filterName “HaplotypeScore”, --filterExpression “HaplotypeScore > 13.0” [37].

For the above candidate SNPs, further filtering was performed if they exhibited the following characteristics: (i) SNPs were located in low-complexity or simple-repeat regions, (ii) the read depth at the variant position was lower than 4 or higher than 50, (iii) as high frequency of false-positives (FPs) occurring around InDels, the adjacent 50 bp of target genomic regions were excluded, and (iv) variant sites located from the gap within 3 bp and adjacent 10 bp sites.

### 2.6. Genotype Inheritance State and Mutation Analysis

“Genotype” refers to both alleles at one position. There is a genotype position in each genome we sequenced that corresponds to each position in the reference genome. As genotype callers, such as GATK, assign diploid genotypes to autosomal loci, regions of heterozygous deletion are erroneously assigned as homozygous genotypes. In the context of triple designs, variants within heterozygous deletions frequently exhibit Mendelian errors as a result of this genotype misassignment [38]. Therefore, we masked these SNPs located in copy number variation (CNV) regions detected by CNVnator [39] and repetitive sequence regions detected by RepeatMasker [40].

Two methods were chosen to analyze genotype transmission patterns in *Equus* progenitor-progeny, and only the intersection of non-Mendelian inheritance loci identified by the two methods were used for subsequent analysis. First, referring to Roach’s method [41], we analyzed the genotype transmission patterns in three family members using bcftools (version 1.3.1, Wellcome Trust Sanger Institute, Wellcome Trust Genome Campus, Cambridge, UK) software and vcftools (version 0.1.16-15, Wellcome Trust Sanger Institute, Wellcome Trust Genome Campus, Cambridge, UK) software. Second, we directly screened SAMtools mpileup data to identify *de novo* SNPs and Mendelian inheritance error (MIE) SNPs using the “trio” command in VarScan (version 2.4.2) software with the following parameters: --min-coverage 10, --min-var-freq 0.20, --*p*-value 0.05, --adj-var-freq 0.05, and --adj-*p*-value 0.15 [42]. These non-Mendelian SNPs were retained, which met the following two requirements: (1) every SNP was supported by no less than ten reads; (2) the read counts of SNPs from the two parental alleles were not fewer than five [43].

### 2.7. Gene Annotation of MIE SNPs and De Novo SNPs

ANNOVAR software was used to annotate the function of SNPs in the mule genome. For SNPs located in intergenic regions, only genes within 5 kb were retained. The clusterProfiler package of R software was used to KEGG enrichment analysis with the following parameters: organism = “ecb”, keyType = “kegg”.

## 3. Results

### 3.1. The Equus Parent-Offspring Trio Genomes

Whole-genome sequences from an *Equus* parent-offspring trio were analyzed. The donkey genome used in this study was stored in our laboratory (Appendix A) [5]. For mule and horse genomes, PE libraries were constructed and sequenced on the Illumina HiSeq X-ten platform (Appendix A). In total, 103.78 Gb, 100.39 Gb, and 114.36 Gb of high-quality genome sequences were generated for the horse, donkey and mule, respectively, after stringent filtering (Appendix A). Approximately equal amounts of data were intercepted to avoid compromising analytical accuracy due to the difference in the original input data. Approximately 97 Gb bases with an average depth coverage of 38× were aligned to the thoroughbred horse reference genome (*EquCab3.0*) (Appendix A).

Unique alignments were retained for subsequent analysis after filtering with mapping quality < 30. In total, 28,272,430, 5,918,968, and 28,287,142 SNPs were identified in the genomes of the donkey, horse and mule, respectively, by both the SAMtools and GATK software programs (Appendix A). These initially obtained candidate sites were further examined to minimize systematic errors and false-positives for the accuracy. Furthermore, we only considered autosomes, for which SNPs were evenly distributed across chromosomes (Appendix A). Of these, there were approximately 24,861,384 positions at which at least one family member had an allele that varied from the reference genome (Appendix A).

As shown in Table 1, we identified 5,012,403 SNPs in the Mongolian horse. This was comparable to the results of previous reports, which reported a range from 3,639,479 to 6,040,778 [44,45]. Heterozygous SNPs in the Mongolian horse were more abundant than homozygous SNPs, which was also consistent with the previous reports above. We identified 23,819,055 SNPs in the domestic donkey (thoroughbred horse genome as reference), and this value was also comparable to previous reports of approximately 24,076,918 SNPs [2]. The frequency of SNPs in the donkey genome (0.9501%) and the mule genome (0.9344%) was considerably higher than that in the horse genome (0.1999%), which could be explained by the identification of SNPs using the thoroughbred horse genome as the reference.

### 3.2. Characterization of SNP Transmission from Parents to Offspring

The purpose of the present study was to identify and analyze the genetic signature of point mutations in the mule genome from the parent-offspring trio. Referring to Roach’s method [41], in this *Equus* trio family, the transmission pattern for each variant position was grouped into three inheritance states. First, the mule received one allele from the horse and the other from the donkey, whose variant position was classified as a Mendelian inheritance SNP (Appendix A). Second, the mule received a pair of alleles from one parent, whose variant position was classified as a Mendelian inheritance error (MIE) SNP (Appendix A). Third, the mule received at least one allele that was not from the parents, whose variant position was classified as a *de novo* SNP (Appendix A).

Of the 24,861,384 variant positions, 14,154,723 were inherited through the Mendelian inheritance pattern in parent-offspring members (Appendix A). A total of 25,703 MIE SNPs transmitted from parental genomes were identified in the mule genome (Figure 1, Appendix A), which is almost 32 times more than the number of natural MIE mutations in chimpanzees (794) from a chimpanzee parent-offspring trio [46]. Compared with parental SNPs, 555 SNPs in the mule genome were characterized as novel, which is almost 12 times more than the natural *de novo* mutations in the chimpanzees (45) mentioned above (Figure 1, Appendix A). The rate of *de novo* SNPs is 2.21 × 10^−7^ in the *Equus* parent-offspring trio. The natural mutation rate for *Equus* is 7.24 × 10^−9^ (per site per generation) [1] and ranges between 0.82 and 1.70 × 10^−8^ in humans [47,48].

### 3.3. Functional Annotation of SNPs

To investigate the potential functions of these SNPs in the mule genome, we searched for genes within approximately 5 kb of the SNPs using ANNOVAR software. A total of 5625 genes were annotated. In total, 11,595 MIE SNPs inheriting from the donkey were embraced by 1453 genes, the other 14,108 MIE SNPs from the horse were embraced by 4557 genes, and 555 *de novo* SNPs were embraced by 658 genes (Appendix A). As shown in Figure 2, a high frequency of SNPs was located in intergenic regions (14,927, 56.85%); this was followed by intronic regions (6506, 24.78%), ncRNA intronic regions (1665, 6.34%), exons (956, 3.64%), upstream regions (673, 2.56%), downstream regions (608, 2.32%), UTR3 regions (386, 1.47%), ncRNA exons (374, 1.42%), and UTR5 regions (149, 0.58%). These findings are consistent with the general distribution reported previously.

Of these, lots of point mutant SNPs were located directly within or adjacent to immune genes, including MHC class I genes (e.g., *MHCX1*, *EQMCE1*) and MHC class II genes (e.g., *DRA*, *DQB*), immunoglobulin family genes (e.g., *LOC100054595*) and various critical cytokines, such as interleukin genes (e.g., *IL-1β*, *IL-12β*), interferon genes (e.g., *IFN-γ*), and intercellular adhesion molecule 1 (*ICAM1*). In the F1 heterozygous mule, genes with mutations in DNA sequences were also highly concentrated in DNA replication (e.g., *PRIM1*, *POLD2*), DNA repair (e.g., *PMS2* and *MSH2*), and cancer, such as proto-oncogenes (e.g., *KRAS*, *HRAS*) and tumor-suppressor genes (e.g., *APC*, *PTEN*).

### 3.4. KEGG Pathway Enrichment Analysis

Pathway analysis showed that the mutant genes were enriched in 344 KEGG pathways (Appendix A). A large number of pathways playing crucial roles in the immune response processes, including antigen processing and presentation, specific antigen recognition, naive lymphocyte activation, proliferation and differentiation, and effector cell generation and production of effects, were enriched (Appendix A). The antigen processing and presentation pathway (ecb04612) was enriched by mutant genes in this study, including MHC class I genes (e.g., *MHCX1*, *EQMCE1*) and MHC class II genes (e.g., *DRA*, *DQB*), by which peptide antigens derived from cytosolic proteins, both self and foreign, are processed for presentation at the cell surface to restricted T lymphocytes and initiate adaptive immune responses (Figure 3) [49]. In addition, endocytosis (ecb04144) and FcγR-mediated phagocytosis (ecb04666), which are essential for antigen presenting cells to recognize and internalize exogenous antigens, were enriched. Numerous pathways involving the subsequent processing of antigens were enriched, including phagosome (ecb04145), lysosome (ecb04142), and the ubiquitin-proteasome degradation process (ecb04120, ecb03050). Through these processes, peptide substrates are degraded into suitable antigenic peptides for presentation to T cells. The T cell receptor signaling pathway (ecb04660, e.g., *FYN*, *GRB2*) was enriched. The essential function of this pathway is to transduce extracellular stimuli into intracellular signals and then initiates signaling cascades, culminating in the response to antigen receptor engagement. The MAPK signaling pathway (ecb04010) and NF-κB signaling pathway (ecb04064), two major downstream pathways of TCR signal propagation, were enriched. The Th1 and Th2 cell differentiation pathway (ecb04658) and Th17 cell differentiation pathway (ecb04659) were enriched, which directly coordinate CD4^+^ T cell differentiation. These differentiation programs are also dynamically regulated by the JAK-STAT signaling pathway (ecb04630). An appropriate cellular energy environment ensures that naive T cells successfully proliferate and differentiate into specific T cell subsets for an effective immune response. Multiple metabolism-related signaling pathways were enriched in this study, such as PI3K-Akt (ecb04151), mTOR (ecb04150), AMPK (ecb04152), and Foxo (ecb04068) pathways, which are required to coordinate T cell metabolic activity [50,51,52,53].

Many mutant genes are directly involved in cell fate processes, such as the cell cycle (ecb04110, e.g., *CCNB1*, *CCND1*, *CDC25A*) and cellular senescence (ecb04218, e.g., *ATM*) pathways. These mutant genes were also involved in the cAMP signaling pathway (ecb04024, e.g., *PDE10A*, *PDE4D*) and the cGMP-PKG signaling pathway (ecb04022, e.g., *PDE5A*, *PDE3A*). These interacting kinases are critical for regulating cell fate by dynamically regulating the concentration of cAMP [54]. The DNA replication pathway (ecb03030, e.g., *PRIM1*, *POLD2*), which regulates the DNA replication process, was enriched. Mutant genes were also enriched in DNA damage responses, including mismatch repair (ecb03430, e.g., *MSH2*), nucleotide excision repair (ecb03420, e.g., *ERCC4*), base excision repair (ecb03410, e.g., *PARP4*), and two major double-strand break repair pathways: non-homologous end-joining (ecb03450) and homologous recombination (ecb03440) (Appendix A) [55,56].

Pathways in cancer (ecb05200) and another 17 cancer- or tumor-related pathways (e.g., colorectal cancer, renal cell carcinoma and pancreatic cancer) were directly enriched by mutant genes in the mule genome (Appendix A). In addition, transcriptional regulation of cancer pathways, such as transcriptional misregulation in cancer (ecb05202, e.g., *MYC*, *MLLT10*) and microRNAs in cancer (ecb05206, e.g., *MIR125B*), were enriched. Several metabolic regulatory pathways participated in cancer, such as the proteoglycans in cancer pathway (ecb05205, e.g., *TWIST2*, *HGF*, *EGFR*), choline metabolism in cancer pathway (ecb05231, e.g., *PLA2G4A*, *PLD1*) and central carbon metabolism in cancer pathway (ecb05230, e.g., *LDHB*, *HK2*), were also enriched. The viral carcinogenesis pathway (ecb05203) and chemical carcinogenesis pathway (ecb05204, ecb05207, ecb05208, e.g., the cytochrome P450 gene family, the glutathione-S-transferase gene family, and the uridine diphosphate-glucuronosyl transferase gene family) were enriched. Many cancer-related pathways, including the wnt (e.g., *NLK*, *DAAM1*) and p53 signaling pathways, were also specifically enriched by mutant genes in the mule [57,58].

## 4. Discussion

In this study, the rates of SNP in the domestic donkey and Mongolian horse are comparable to previous reports [2,44,45]. Referring to the reports of Tatsumoto [46] and Roach [41], point mutations, including MIE SNPs and *de novo* SNPs, were identified in the mule genome through the whole-genome sequences of an *Equus* parent-offspring trio. The mule has a higher mutation rate than the natural mutation rate in chimpanzees and humans, as well as in *Equus* [1,46,47], which is consistent with the hypothesis tested experimentally by Duncan in 1915 that interracial crosses may have a high mutation rate [59]. A limited number of plant genome-wide studies also indicated that heterozygotes have higher mutation rates than homozygotes [29,30]. We cannot deny that there were incorrectly identified SNPs due to limited sequencing data, although the mutation rate in mules conformed to the rule that heterozygotes have higher mutation rate.

In previous reports by Liu and his colleagues, they identified 12,146 (the genomes of goldfish as reference,) and 58,587 (the genomes of common carp as reference) offspring-specific mutations in the hybrid fish transcriptome [28]. The mutation rate in hybrid fish is obviously higher than that observed in mules. However, we observed that the mutation rate in the mule was ten times higher than the one observed in heterozygous plants [30]. That is contrary to the previous conclusion that plants have higher point mutation rates than animals, depending on the transgenic tissue analyzed [60]. Mutation rates vary among species, which may be attributed to the species itself or the length of separation time. These results were generated by a few studies on a limited number of experimental samples. In the future, we need to further verify these conclusions by relying on more samples.

In summary of previous reports [48,61,62], the origin of these point mutations may be as follows: (i) germline events induced by parental meiosis during gametogenesis, (ii) postzygotic events that occurring in offspring mitosis early during embryogenesis, or (iii) specific mutations in blood cells of the offspring after tissue differentiation. Our working speculates that the higher mutation rate in the mule genome is attributable to rapid post-zygotic somatic mutation triggered by shock stress integrating the horse and donkey haploids into the mule genome. The molecular mechanism of the high mutation rate in mule genome may be the defect of DNA mismatch repair caused by sequence divergence in horses and donkeys [63], but the data in this study is not enough to explain the detailed mechanism. According to the above hypothesis, these point mutations in mule may reflect the incompatible loci in mutational differences accumulated between the horse and the donkey. We also speculate that the rapidly mutating SNP may act as a buffer to balance the incompatibility of parental haploid genomes and thus protect the survival of mule.

*MHC* genes play critical roles in maintaining appropriate immune homeostasis and self-tolerance, and are central to the vertebrate immune response, which is necessary for health. Similarly to most mammals, *Equus MHC* genes are characterized as extreme polymorphisms and trans-species polymorphisms that are maintained by pathogen-driven balanced selection, and lineage-specific alleles in the antigen binding site (ABS) facilitated by geographic subdivision between horses and donkeys [64,65,66]. This genetic heterozygosity may reduce the fitness, survival, or reproductive success of mules, as reported in a study on the crossbreeding of mice from natural populations by Petteri [17]. Genetic data from various species demonstrate that immunoglobulin loci driven by the diversity of target microorganisms for IG responses are not conserved across the closely related sister taxa. The uncoordinated evolution of heavy- and light-chain gene sets of IGs between the horse and donkey may result in poor interactions in the mule, ultimately reducing fitness, as illustrated in a study using hybrid F1 mice by Watson [21].

It is possible that genes of the immune system have played important roles in interspecific hybridization incompatibility. Autoimmune-like responses induced by DMI have been described in hybrid necrosis in plants [67]. These incompatibilities occur when gene divergence affects loci encoding interacting products, such as receptors and their ligands. During negative selection in the thymus, the strength of the pMHC:TCR interaction is higher than the threshold required for negative selection due to MHC polymorphisms located at or near the peptide-binding groove, resulting in failing to trigger depletion of autoreactive T cells in the medulla [68]. Polymorphisms at MHC class II (e.g., *DQA*, *DQB*, *DRB*) loci account for a higher risk for all autoimmune diseases, such as rheumatoid arthritis [69], type I diabetes mellitus [70] and systemic lupus erythematosus [71], than any other loci in the genome. The impeccable performance of immune responses and immune homeostasis reflects the accurate delivery of signals from external stimuli, which is affected by various factors, such as the intensity and duration of TCR or BCR signaling, the strength of downstream cascade signaling, the specificity of the binding cytokine with its receptor and appropriate gene expression regulated by transcription factors [72]. ICAM1 cooperates with TCR-pMHC complexes to form immune synapses and generate costimulatory signals in response to antigenic stimulation [73]. Polymorphisms of the *ICAM1* gene also play a crucial role in the pathological process of rheumatoid arthritis [74]. TGF-β and IL-1β have the capacity to instruct naive T cells to develop into IL-17-producing Th17 cells which trigger inflammatory responses, including neutropenia, tissue remodeling, and the production of antimicrobial proteins [75]. As previously reported, SNPs in the proinflammatory cytokine genes *IL-1β* and *IFN-γ* are associated with susceptibility to rheumatoid arthritis [76]. SNPs in interleukin *IL-12β* is associated with susceptibility to inflammatory bowel diseases [77], and SNPs in anti-inflammatory *IL-4*/*IL-4R* is associated with susceptibility to type I diabetes mellitus [78].

Mutations might change the enzyme activity or fidelity by affecting standard transcriptional processing. The accumulation of mutational differences in critical kinases that regulate the cell cycle, DNA replication, and DNA repair between horses and donkeys may alter enzyme specificity in mules. Ultimately, this would severely restrict cell fate, manifesting as genomic instability, apoptosis or carcinogenesis. Periodic and phase-specific cyclin expression regulates the cell cycle by activating cyclin-dependent kinases. Overexpression of the cyclin D1 (*CCND1*) gene leads to uncontrolled cell cycle regulation and abnormal cell proliferation, eventually resulting in tumorigenesis [79]. It is well-accepted that *PRIM1* is responsible for DNA replication initiation by synthesizing short RNA-DNA primers, and *POLD*, a high-fidelity DNA polymerase, is used for processive elongation on the leading and lagging strands of DNA [80,81]. DNA polymerase also plays a crucial role in DNA repair. DNA polymerase inefficiency may cause detrimental consequences, such as chromosomal instability and oncogene activation, which lead to carcinogenesis. Previous reports by Ranga et al. in mice indicated that heterozygous mutations at the polymerase active site of DNA polymerase reduces lifespan, increases genomic instability, and accelerates tumorigenesis in an allele-specific manner [82]. As implied by Lynch in the *MLH1* (also known as *PMS2*)-deficient Baker’s yeast strain, DNA mismatch repair genes repair over 90% of replication errors [83]. The heterozygosity of enzymes, integrating the divergent parental haploids, may alter enzyme efficacy, which may also be a reason for the high frequency of mutations in mule.

A complex regulatory network integrating gene expression, energy metabolism, and extracellular matrix controls cell fate. However, dysregulation of this network remodeling can lead to tumorigenesis and cancer development by providing favorable conditions for tumor cells [84,85]. Mutations in proto-oncogenes (e.g., *KRAS*, *HRAS*) and tumor-suppressor genes (e.g., *APC*, *PTEN*) are the primary drivers in various cancers [86,87,88]. Altered gene expression is another primary molecular mechanism of cancer pathology. The *MYC* gene and the *MIR125B* microRNA are directly involved in establishing the specific programs of gene expression in cancer cells, comprising almost all aspects of cancer biology, such as proliferation, apoptosis, invasion/metastasis, and angiogenesis [85,89]. The proteoglycans in cancer pathway (ecb05205) regulates proteoglycans in the extracellular matrix, which influences the behavior of cancer cells and their microenvironment by interacting with various cytokines (e.g., *TWIST2*), growth factors (e.g., *HGF*) and cell surface receptors (e.g., *EGFR*) [90]. The choline metabolism in cancer pathway (ecb05231) leads to increased levels of choline-containing precursors via the modulation of enzymes (e.g., *PLA2G4A*, *PLD1*) [91]. The central carbon metabolism in cancer pathway (ecb05230, e.g., *LDHB*, *HK2*) controls cancer metabolic adaptations through aerobic glycolysis, elevated glutaminolysis, dysregulated tricarboxylic acid cycle and pentose phosphate pathways [92]. Chemical compound exposure and viral infection are two common exogenous factors responsible for carcinogenesis. In general, chemical carcinogens require metabolic activation to produce reactive intermediates capable of binding to cellular macromolecules, which are necessary for chemicals to cause cancer. The cytochrome P450 gene family, the glutathione-S-transferase gene family, and the uridine diphosphate-glucuronosyl transferase gene family are critical enzymes responsible for the metabolism of chemical carcinogens [93].

## Figures and Tables

**Figure 1 genes-13-02188-f001:**
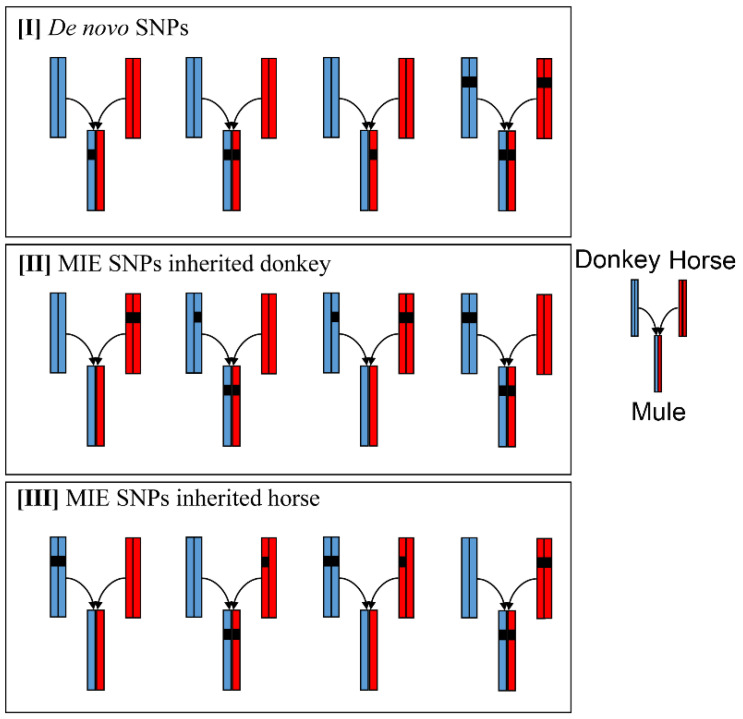
Analysis of non-Mendelian inheritance SNPs. Classification of non-Mendelian inheritance SNPs. When variant alleles were identified only in the mule, they were classified as [I] *de novo* SNPs. MIE SNPs were classified into [II] inherited from the donkey and [III] inherited from the horse.

**Figure 2 genes-13-02188-f002:**
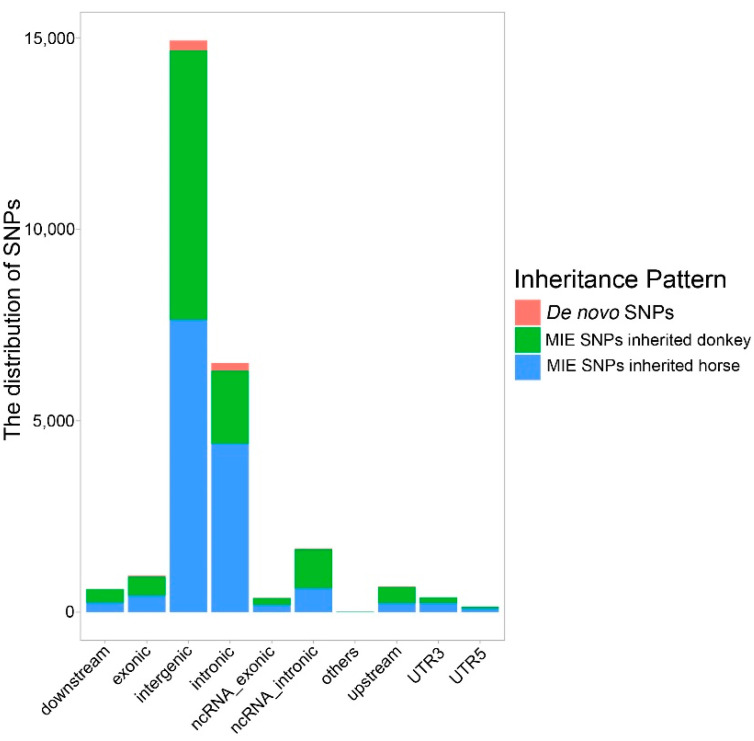
The statistics of SNP distribution.

**Figure 3 genes-13-02188-f003:**
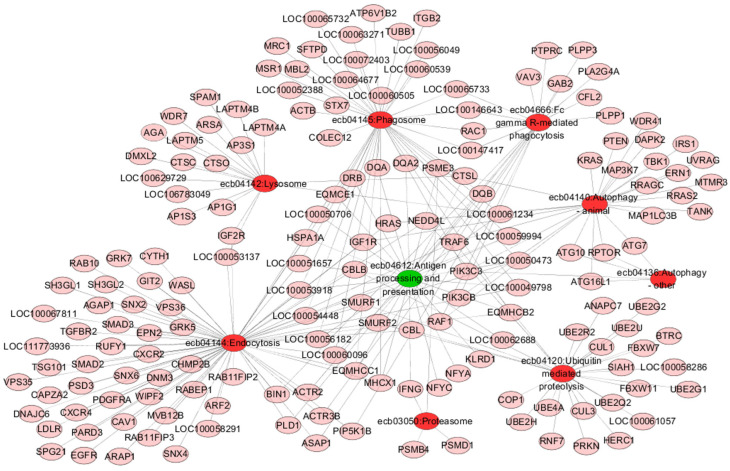
Integrated pathway network analysis of point mutation genes involved in antigen processing and presentation in the mule genome. Green and red represent pathways; pink represents genes.

**Table 1 genes-13-02188-t001:** Summary of SNPs aligning to the horse reference genome (*EquCab3.0*).

Samples	Donkey	Horse	Mule
Depth	4 ≤ depth ≤ 50	4 ≤ depth ≤ 50	4 ≤ depth ≤ 50
Heter. SNPs	1,996,879	3,387,403	21,771,865
Homo. SNPs	21,822,176	1,625,000	1,654,376
Total SNPs	23,819,055	5,012,403	23,426,241
%SNP	0.950115	0.199939	0.934446
% Heterozygosity	0.0797	0.135	0.868
Transitions	16,302,515	3,386,154	16,029,003
Transversions	7,516,540	1,626,249	7,397,238
Ti/Tv (autosome)	2.17	2.08	2.17

## Data Availability

The genome data were submitted to NCBI as project accession PRJNA842856 (SRA accession: SRR19427107, SRR19427108) for the horse and mule. The remaining data are available within the article and its Appendix A files or available from the authors upon request.

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
