# Peer review of "Analysis of the Whole-Genome Sequences from an Equus Parent-Offspring Trio Provides Insight into the Genomic Incompatibilities in the Hybrid Mule"

_genes, 2022, doi:10.3390/genes13122188_

Round 1
Reviewer 1 Report
This is a thorough and well-referenced manuscript with intriguing findings about genomic polymorphisms in hybrid offspring. The topic of polymorphism and genomic incompatibilities is not extensively studied and mules present a natural model, particularly when used as a parent-offspring trio.
The methods and results are described well, with ample supplemental materials and data shared publicly. However, the following items should be revised prior to publication:
1. Add a p-value to the abstract following the words "significantly higher" (line 22).
2. Line 95 states that there is 97.5% similarity between horse and donkey genomes and refers to Table S1, however that table does not provide similarity values. Please correct this.
3. The meaning of sentences on lines 75-77 and 88-93 is unclear. These need revision for clarity. The word "whether" seems to be misunderstood.
4. The sentence on lines 41-42 needs a reference.
5. References 81-85 need to be re-ordered based on the order they are used in the text.
6. "Illumina" is mis-spelled as "Illumine" in the abstract and on line 100, please correct.
Reviewer 2 Report
In this study, Ren et al. generated genome-wide data from a interspecific parent-offspring trio, and identified a surprisingly high de novo mutation rate in the mule offspring, 40x higher than the natural mutation rate estimated for the horse. Although the analyses seem sound and follow the standards in genome mapping and SNP identification, I have several major concerns about the interpretation of these results as well as the impact and significance of the study. In particular, while this extremely high mutation rate may be true, I believe additional analyses are needed to strongly support this finding.
First, I am surprised that the authors do not comment more on this massive mutation rate, and that they are not concerned about such a high number. I highly recommend some additional filtering procedures to be increase confidence in the true de novo nature of the mutations identified. Here are two such examples:
- Increase the minimum read depth coverage from 4 to 5, and to 10, since more than 97% of all mule mapped data has a coverage higher than 10% (Table S6). I would not be surprised that many false positives arose from sequencing errors at sites where coverage is low. As mentioned below more in detail, I kindly ask the authors to show a distribution of read coverage for the SNPs identified, and for the de novo ones specifically.
- An easy way to be confident about the identified SNPs is to compare them with a panel of horses and donkeys whose genomes have been sequenced. This approach is based on the rationale that the probability that these point mutations occur at positions that are already variable in the horse and/or the donkey populations is extremely low. Any de novo SNP segregating at positions that are variable in the population is most likely spurious and should be filtered out. Since dozens of horse and donkey genomes are publicly available, I strongly urge the authors to compare them to the mule genome they have generated here.
Second, there is no mention in methods section of the parameters used to functionally annotate SNPs, perform KEGG pathway enrichment analysis. This is problematic as there is no mention of significance thresholds used to determine enrichment in specific pathways/categories, and no mention of the gene universe chosen to conduct such analyses either.
Third, a more minor point that may turn out to be very important: the authors do not comment on the differences in SNPs identified between SAMtools and GATK. Why do such differences arise? I would be very interested in reading the opinion of the authors on this matter. In figure S2, 4-5% of all SNPs identified are found by only one of the two algorithms. Similarly, in figure S3, 25%, 39% and 33% of all identified inDels are detected by only one of the two algorithms.
Finally, and this is in my opinion a major concern that negatively impacts the conclusions of the paper, the text is very difficult to read due to many issues in grammar and syntax, all along the manuscript. There are many aspects of the reasoning that I simply do not understand, and I would strongly advise the authors to have their manuscript proof-read.
Some other points
l.39-41 ‘the mule (…) and the hinny (…) are scientifically considered sterile’. Several cases of fertile female mules have been observed. Please edit this sentence accordingly.
l.41-42: ‘pregnant mares giving birth to mules are more likely to miscarry than those giving birth to normal foals’: please cite relevant literature to back up this bold statement.
l.47-48. The authors comment on fitness reduction in hybrids, but what about heterosis and enhanced hybrid fitness? There is a reason the mule was extensively used in Roman antiquity…
l.59-60. MHC and IG are ‘characterized as extreme polymorphism maintained by pathogen-driven balanced selection’: please cite relevant literature to back up this bold statement.
l.94. ‘data saved in our laboratory’: I am very confused by this statement, what do the authors mean exactly?
l.131. I am curious to know whether the authors have tried mapping the data to the genome of the donkey. If not, I would strongly encourage them to do so, and then compare the differences obtained from mapping to the horse vs to the donkey.
l.154: I am surprised that the authors chose such a low minimum read depth. With a depth coverage of 4, sequencing errors can still have a strong impact on the observed genotype.
To make sure this is not an issue creating false positives, I kindly ask the authors to show the coverage distribution of de novo SNPs they have identified.
l.176. ‘MIE’: please explicitly write what MIE stands for.
l.182-184: ‘For mule and horse genomes, PE libraries (400 bp) were constructed and sequenced on the Illumina Hiseq X-ten platform’: this is redundant with the methods section and should be removed.
l.330-332: when highlighting similarities in higher mutation rate in heteroyzgotes than in homozygotes, the authors omit to mention that the mutation rate they observe in the mule here is 10 times higher than the one observed in plants. Such a high mutation rate should raise concerns and be addressed in the discussion.
Figure S1B: Why does the read depth coverage distribution look bimodal? This is surprising and a bit concerning to me.
Round 2
Reviewer 2 Report
The authors have provided a revised version of their manuscript and have tried to address all of the problematic points. While I am please with their responses to the minor points, I have a major concern about the total number of De Novo mutations the authors have found, and all subsequent analyses in their manuscript.
”Response: The distribution of read coverage for MIE SNPs and de novo SNPs identified has been shown in Table S10, and de novo SNPs with depth coverage ≥10 can account for 97.23%.”
I scrutinized Table S10 and the distribution of depth coverage, and I must say that several things stroke me and make me think that the number of de novo mutations identified is spurious and cannot be trustworthy. First, in the list of De Novo mutations identified in the mule, 195 have a reference coverage £ 10 and 443 have an alternative coverage £ 10. With such low coverages, how can you be sure that these are true de novo mutations? Sites with a low reference coverage are very likely to be spurious and not informative at all because of (1) sequencing errors, and more importantly, (2) reference bias. Similarly, sites with a low alternative coverage may simply be false positives, again because of sequencing errors.
I had a closer look at other papers estimating de novo mutations have set way more conservative thresholds. For example, I invite the authors to read carefully Tatsumoto et al. 2017, in which they downsample genome-wide data to show that the vast majority of de novo mutations identified with low-coverage data (30X) are not found with higher coverage threshold (<30X). They even emphasize than 90X may not be enough to accurately call de novo mutations. Tatsumoto and colleagues write “deep-sequencing coverage data for all the members are important to call variants at heterozygous sites reliably and to identify de novo SNVs with minimum FPs and FNs”.
In the light of the gold standards in detecting De Novo mutations, I have trouble understanding why the authors chose to keep sites with so low read depths, and why they did not sequence more. I would advise the authors to generate additional sequencing data and put a much higher threshold for minimum read depth, in any case higher than 30X.
The high proportion of false positives among identified de novo mutations is also hinted by their distribution of read depth that I inferred from the data in Table S10 (attached pdf). Granted that this graph is a very draft one, it still shows a distribution that is very different from that of whole-genome data! In particular, it is not normal, the average is way lower than for the whole-genome data (<27X vs 46X for whole-genome data!), and only 26 of these sites have a coverage higher than the overall mean value for whole-genome data.
Overall, the literature and the data shown in this manuscript strongly suggest that the vast majority of identified de novo mutations are actually false positives, and call for (1) more sequencing, (2) more stringent thresholds.
“An easy way to be confident about the identified SNPs is to compare them with a panel of horses and donkeys whose genomes have been sequenced. This approach is based on the rationale that the probability that these point mutations occur at positions that are already variable in the horse and/or the donkey populations is extremely low. Any de novo SNP segregating at positions that are variable in the population is most likely spurious and should be filtered out. Since dozens of horse and donkey genomes are publicly available, I strongly urge the authors to compare them to the mule genome they have generated here.”
The authors did not comment on this point in their response. This is crucial if the authors want to ascertain that the mutations identified are not false positives. The rationale behind this is very simple: if some of the de novo mutations are found to be present at a reasonable frequency in donkey genomes or horse genomes, it is highly likely that they are false positives, since the odds of having a SNP specifically at the same position in the genome are extremely low. I cannot insist enough on how important applying such a check is to guarantee that the mutations identified are true ones and not artefacts.
”Response: Because the donkey genome data used in this study was intercepted from the donkey reference genome data, we did not use the donkey genome as the reference genome.”
I do not understand this response point. Have the authors generated sequencing data or have they been using the donkey reference genome? Alternatively, have they been generating more sequencing data from Willy, the donkey that serves as reference genome for this species? If they have generated data, surely they have raw sequencing data, in the form of fastq files, which can be mapped to the donkey genome?
This is in my opinion extremely important: de novo mutations must be identified from both data mapped to the horse genome and data mapped to the donkey genome to be confident that these mutations are genuine!
“Response: We once again used fastp software to filter the raw data and then mapped the clean data to the reference genome. However, we obtained similar depth coverage results. In addition, it is worth mentioning that 99.17% of the raw data is high-quality after quality filtering. This indicates that the raw data generated by sequencing is reliable. “
Having high-quality data after filtering is not enough in this case. Any bimodal distribution of reads should be of concern, and further investigated. How can the authors explain that coverage between ~35X and 50X is much higher than expected under a normal distribution of read depth? Please comment on this, and map the data to the donkey genome to see whether this pattern still holds true.
